# Additive manufacturing of 3D nano-architected metals

Andrey Vyatskikh [1], Stéphane Delalande[2], Akira Kudo[1], Xuan Zhang[3], Carlos M. Portela[1] & Julia R. Greer[1]

Most existing methods for additive manufacturing (AM) of metals are inherently limited to ~20–50 μm resolution, which makes them untenable for generating complex 3D-printed metallic structures with smaller features. We developed a lithography-based process to create complex 3D nano-architected metals with ~100 nm resolution. We first synthesize hybrid organic–inorganic materials that contain Ni clusters to produce a metal-rich photo-resist, then use two-photon lithography to sculpt 3D polymer scaffolds, and pyrolyze them to volatilize the organics, which produces a >90 wt% Ni-containing architecture. We demonstrate nanolattices with octet geometries, 2 μm unit cells and 300–400-nm diameter beams made of 20-nm grained nanocrystalline, nanoporous Ni. Nanomechanical experiments reveal their specific strength to be 2.1–7.2 MPa $g^{-1}$ $cm^3$, which is comparable to lattice architectures fabricated using existing metal AM processes. This work demonstrates an efficient pathway to 3D-print micro-architected and nano-architected metals with sub-micron resolution.

[1] Division of Engineering and Applied Sciences, California Institute of Technology, 1200 E. California Blvd., Pasadena, CA 91125, USA. [2] Scientific Department, PSA Group, Centre Technique de Vélizy 2, route de Gizy, Vélizy-Villacoublay 78943, France. [3] Center of Advanced Mechanics and Materials, Department of Engineering Mechanics, Tsinghua University, Beijing 10084, China. Correspondence and requests for materials should be addressed to J.R.G. (email: jrgreer@caltech.edu)

Additive manufacturing (AM) represents a set of processes that enable layer by layer fabrication of complex 3D structures using a wide range of materials that include ceramics[1], polymers[2], and metals[3]. The development of metal AM has revolutionized the production of complex parts for aerospace, automotive and medical applications[4,5]. Today's resolution of most commercially available metal AM processes is ~20–50 μm[6]; no established method is available for printing 3D features below these dimensions[7]. It has been shown that unique phenomena arise in metals with micro-dimensions and nano-dimensions, for example light trapping in optical meta-materials[8] and enhanced mechanical resilience[9–15]. Accessing these phenomena requires developing a process to fabricate 3D metallic architectures with macroscopic overall dimensions and individual constituents in the sub-micron regime.

Minimum feature size in metal AM is generally limited by the material feedstock, i.e., the method of supplying metal in powder, wire, sheet or ink form during fabrication. Inkjet-based methods[16,17] manipulate 40–60 μm droplets of metal inks, limiting the smallest features to at least the size of a solidified droplet. Wire-based and filament-based processes, such as plasma deposition[4] and electron beam freeform fabrication (EBF3)[18], rely on locally melting a >100 μm-diameter metal wire, which produces millimeter-sized features. Powder-based processes, such as selective laser melting (SLM) and laser engineered net shaping[19], consolidate ~0.3–10 μm metal powder particles, which limits the smallest feature size to about 20 μm[6,20]. Overcoming these resolution limitations requires a capability to manipulate nanoscale quantities of metals in a stable and scalable 3D printing process.

Alternative material feeds to fabricate 3D metal structures with <10 μm resolution include nanoparticle inks, ion solutions, droplets of molten metal, and precursor gases[7]. Methods that use localized electroplating[21,22] or metal ion reduction[23,24] are capable of producing features down to 500 nm using a very slow process that is limited by electroplating rate. Electrochemical fabrication (EFAB) allows for manufacturing geometries with 10-μm features and 4-μm layers, but is limited to structures with a total height of 25–50 layers[25]. Other technologies, like micro-deposition of metal nanoparticle inks[26–28] or molten metal[29] and focused ion beam direct writing, also suffer from slow throughput and are more suited for low-volume fabrication and repair[30].

We demonstrate a facile and reproducible process to create complex 3D metal geometries with a resolution of 25–100-nm. We synthesize hybrid organic–inorganic materials that contain Ni clusters and use them to produce a metal-rich photoresist. We then use two-photon lithography (TPL) to sculpt computer-designed architectures out of the resist and pyrolyze them first in inert atmosphere at 1000 °C and then in reducing atmosphere at 600 °C to volatilize the organic constituents. Using this approach, we demonstrate the fabrication of periodic Ni octet nanolattices with the unit cell size of 2 μm and beam diameters of 300–400 nm diameter as a proof-of-concept. TEM analysis reveals that the microstructure of Ni beams is nanocrystalline and nanoporous, with a 20 nm mean grain size and 10–30% porosity within each beam. Nanomechanical experiments demonstrate that the strength of these Ni nanolattices is comparable to that of the metal lattices with 0.1–1.0 mm beam diameters fabricated using alternative metal AM technologies. These findings suggest an efficient pathway to create complex 3D metal structures with nano-scale resolution.

## Results

### AM of nickel nano-architectures.
We first synthesized nickel acrylate using a ligand exchange reaction between nickel alkoxide and acrylic acid (Fig. 1a) and combined it with another acrylic monomer, pentaerythritol triacrylate, and a photoinitiator, 7-diethylamino-3-thenoylcoumarin (Fig. 1b). We then drop cast this photoresist on silicon substrate and used TPL to sculpt the prescribed 3D architectures (Fig. 1c). The non-polymerized resist was then washed away, and the free-standing cross-linked polymer nano-architectures were then pyrolyzed to volatilize the organic content. This process yielded a replica of the original 3D structure with ~80% smaller linear dimensions made entirely out of metal (Fig. 1d).

We demonstrate the feasibility and efficiency of this methodology by first fabricating nanolattices with 10 μm octet unit cells comprised of 2-μm diameter circular beams out of the synthesized photoresist using layer-by-layer TPL with 150 nm layer thickness. Scanning electron microscopy (SEM) images in Fig. 1f–h reveal that these nanolattices had fully dense beams and uniformly sized, high-fidelity features. Each sample had four unit cells per side, 40 μm, and a height of three unit cells, 30 μm, and was supported by vertical springs at each corner and by a vertical pillar in the center. These supports served as pedestals that would allow the sample to release from substrate after undergoing an isotropic ~80% shrinkage during pyrolysis (see Supplementary Fig. 1).

Pyrolysis was performed in a tube furnace via a two-step procedure: first at 1000 °C in argon to remove most of the organic content from the samples and to consolidate the Ni metal clusters into denser features, which is accompanied by ~5× linear shrinkage in feature size; and second at 600 °C in forming gas, to reduce the oxygen content in the mostly-Ni samples and to facilitate grain growth. SEM images in Fig. 1(i, j) show a representative 3D Ni architecture and convey that the 10-μm unit cells and 2-μm diameter beams in the original polymer-metal structure shrank to produce ~2 μm unit cells and ~300–400 nm diameter beams in the nickel nanolattice. This also implies that 150-nm layer thickness in the polymer structure corresponds to 30-nm layer thickness in the metal structure. The zoomed-in image in Fig. 1j shows that the metal beams are ~10–30% porous caused by pyrolysis.

**Microstructure and chemical composition of as-fabricated metallic 3D architectures.** Chemical composition of the as-fabricated Ni architectures was characterized using energy-dispersive X-ray spectroscopy (EDS), for which we fabricated individual unit cells with tetrakaidecahedron geometries using the same methodology. Figure 2a shows that these structures shrunk from 20-μm wide unit cells and 2-μm diameter beams on 6-μm pillar supports to 4-μm unit cells and 0.4-μm diameter beams after pyrolysis (Fig. 2b). EDS spectrum (Fig. 2d) taken from a beam section shown in Fig. 2c reveals the chemical composition to be 91.8 wt% Ni, 5.0 wt% O, and 3.2 wt% C. A Si peak from the substrate is also present. EDS maps in Fig. 2e–h convey a relatively homogeneous distribution of each element within the printed structure, which consists mostly of nickel metal and is not segregated into individual nickel-rich, carbon-rich, or oxygen-rich phases.

We also fabricated some few-micron long, 25–100-nm diameter metal beams that spanned the 1.25-μm wide opening in a silicon nitride membrane directly on the transmission electron microscopy (TEM) grids (Fig. 3a) to analyze the atomic-level microstructure of pyrolyzed materials. Figure 3b displays a bright-field TEM image taken along a beam that reveals multiple coalesced grains with a mean size of 21.4 ± 2.0 nm (see Supplementary Table 1). The electron diffraction pattern (Fig. 3d) taken from the region shown in Fig. 3c conveys a strong Ni signal and a much weaker contribution from NiO. A representative high-resolution TEM (HRTEM) image (Fig. 3e) of the beam edge contains multiple lattice fringes, which allowed the calculation of

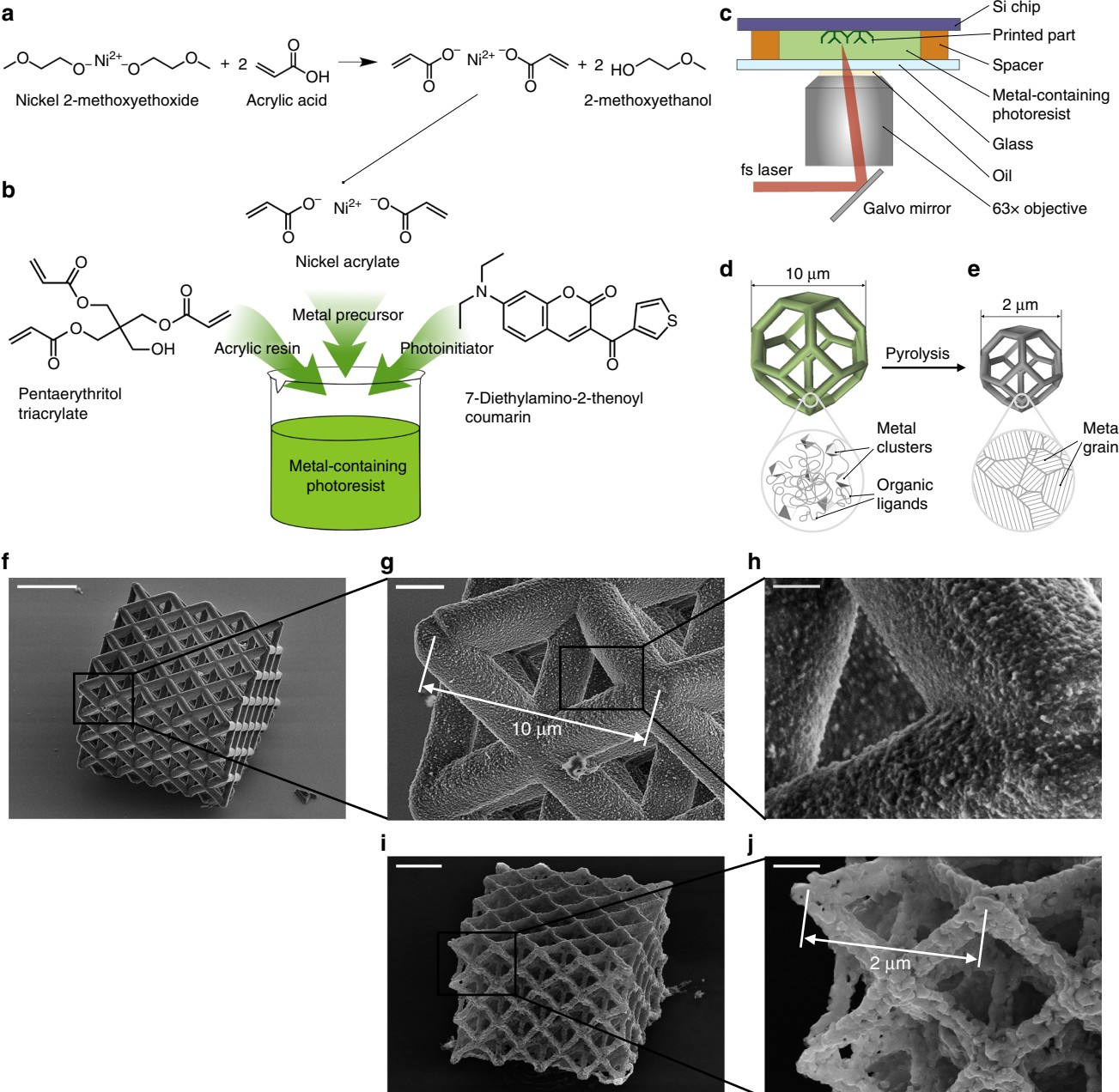

**Fig. 1** Process for nanoscale additive manufacturing of metals and SEM characterization of the fabricated samples. **a** Ligand exchange reaction used to synthesize metal precursor with cross-linking functionality. **b** Metal precursor, acrylic resin, and photoinitiator are mixed to form a transparent metal-containing photoresist. **c** Schematic of two-photon lithography (TPL) process used to sculpt the scaffold. **d** Schematic of fabrication of metal-containing polymer part that is **e** pyrolyzed to remove organic content and to convert the polymer into a metal. SEM images of **f**–**h** a representative octet lattice made out of a nickel-containing polymer at different magnifications and **i,j** a representative nickel nanolattice after pyrolysis. Magnifications in **g** and **i** (scale bars 2 μm) and also **h** and **j** (scale bars 500 nm) are identical. Scale bar is 15 μm for **f**

interplanar atomic spacings using fast Fourier transform (FFT). We identified three distinct spacings: Ni crystals (region 1, spacings of 2.01 and 2.04 Å), $Ni_3C$ particles (region 2, spacings of 1.98 and 2.14 Å), and NiO crystals (region 3, spacing of 2.06 Å). More details can be found in the Methods section. Bright-field TEM revealed that Ni crystals occupy >90% of the examined volume, NiO <10%, and $Ni_3C$ <1%, consistent with EDS results (Fig. 2d and Supplementary Fig. 3). TEM analysis further revealed the presence of nickel (II) oxide nanoparticles with diameter of <5 nm at the surface that were likely formed through surface oxidation in air after TEM sample preparation. Our pyrolysis is equivalent to carbothermal reduction at 1000 °C followed by

reduction by hydrogen and carbon at 600 °C, with no oxygen present in the flowing gas. Literature on this type of thermal treatment reported the composition to be mainly metallic nickel with a minor amount of nickel carbide and/or carbon[31].

**In situ compression of nickel nanolattices.** We conducted uniaxial compression experiments on ten Ni octet nanolattices with relative densities of 27–42% and beam sizes of 300–400 nm (see Supplementary Table 2). The experiments were conducted in situ, in a SEM-based nanomechanical instrument, comprised of a nanoindenter-like module (Nanomechanics, Inc.) inside of SEM

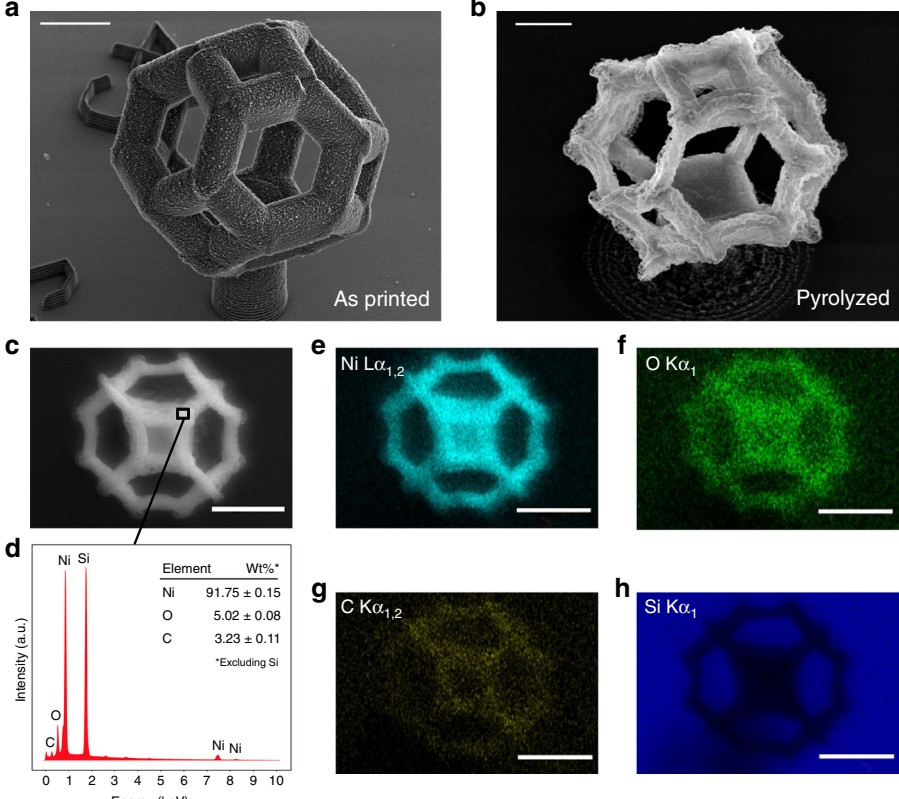

**Fig. 2** Energy dispersive spectroscopy (EDS) characterization of fabricated metal nanostructures. **a** SEM images of supported 20 μm tetrakaidekahedron unit cell on a Si chip before pyrolysis and **b** the same structure after pyrolysis (4 μm width). **c** SEM image of the structure showing where EDS data was collected. **d** EDS spectrum taken from an internal beam region reveals chemical composition to be more than 90 wt% nickel. **e–h** EDS maps show high uniformity of the atomic composition throughout the structure. Scale bars are 5 μm for **a**, 1 μm for **b** and 2 μm for **c**, **e–h**

chamber (Quanta 200 FEG, FEI), which enabled observing the deformation while simultaneously collecting load vs. displacement data[32] (see Supplementary Movie 1). The collected data were converted into engineering stresses and strains by dividing the load by the sample footprint area and dividing the displacement by the initial sample height, respectively. Figure 4a–d shows SEM snapshots obtained during a compression experiment of a representative sample; stress vs. strain data for four representative samples are shown in Fig. 4e (data for additional six samples are presented in Supplementary Fig. 4). All stress–strain data appear to be self-consistent and reproducible. A toe region in the initial portion of each experiment (not shown) is representative of deformation before establishing full contact between the sample and flat punch indenter tip (see Supplementary Fig. 5 for full stress–strain data). The toe region also included the failure of the supporting pillar, which allowed for establishing full contact between the sample and the substrate.

We found that the stress vs. strain data was typical for cellular solids compressions, with the characteristic elastic loading, plateau, and densification sections[33]. The arrows on the plot are correlated with the images above and demarcate specific stages during compression: initial contact (region A), elastic deformation (region B), layer-by-layer collapse (region C), and densification (region D). The point of full contact was determined using harmonic contact stiffness and SEM video. The slope of the elastic loading segment, up to 10–15% strain (region B), was used to estimate structural stiffness of the nanolattices, which ranged from ~47 to 174 MPa. The strength of Ni nanolattices was defined as the maximum stress prior to the first buckling event, marked by open circles in the data in Fig. 4e, and ranged from 6.9 to 18.2 MPa. The elastic region was followed by layer-by-layer collapse

up to 65% strain (region C); two of the four samples were unloaded at 30 and 60% strain. The two other samples were compressed to 70–85% strains, reached densification (region D) and then unloaded (see Supplementary Movie 1). None of the nanolattices recovered after deformation.

## Discussion
EDS analysis revealed that the fabricated nanolattices have a composition of 91.8 wt% Ni, 5.0 wt% O, and 3.2 wt% C. It is reasonable to expect traces of carbon in the pyrolyzed structures caused by the high solubility of carbon in Ni at 1000 °C[34], which leads to carbon precipitation at nickel surface upon cooling down to room temperature. TEM analysis revealed that the carbon also exists in the form of 5 nm-sized $Ni_3C$ precipitates within the beams (Fig. 3e). The accuracy of EDS in quantifying the carbon content may not be sufficient because it is sensitive to the spurious carbon deposited in the SEM chamber[35]. The presence of 5.0 wt% O in the nanolattice can be attributed to formation of a native oxide on Ni surface and to full oxidation of small (<6 nm) Ni surface nanoparticles[36].

Figure 4f shows the specific strength of Ni nanolattices fabricated in this work and those of the metallic lattices fabricated using other metal AM processes as a function of beam diameter on a log–log plot (see Supplementary Table 2 for details). This plot reveals that the specific strength of metallic lattices in refs. [16,37–41] decreases by a factor of 280 as the beam diameter is reduced from 1.78 to 0.04 mm, with the lowest reported specific strength of 0.7 MPa $g^{-1}$ $cm^3$ for octahedral silver lattices. Nanocrystalline Ni nanolattices in this work have the specific strength of 2.1–7.2 MPa $g^{-1}$ $cm^3$, which is ~2–10 times higher than that of

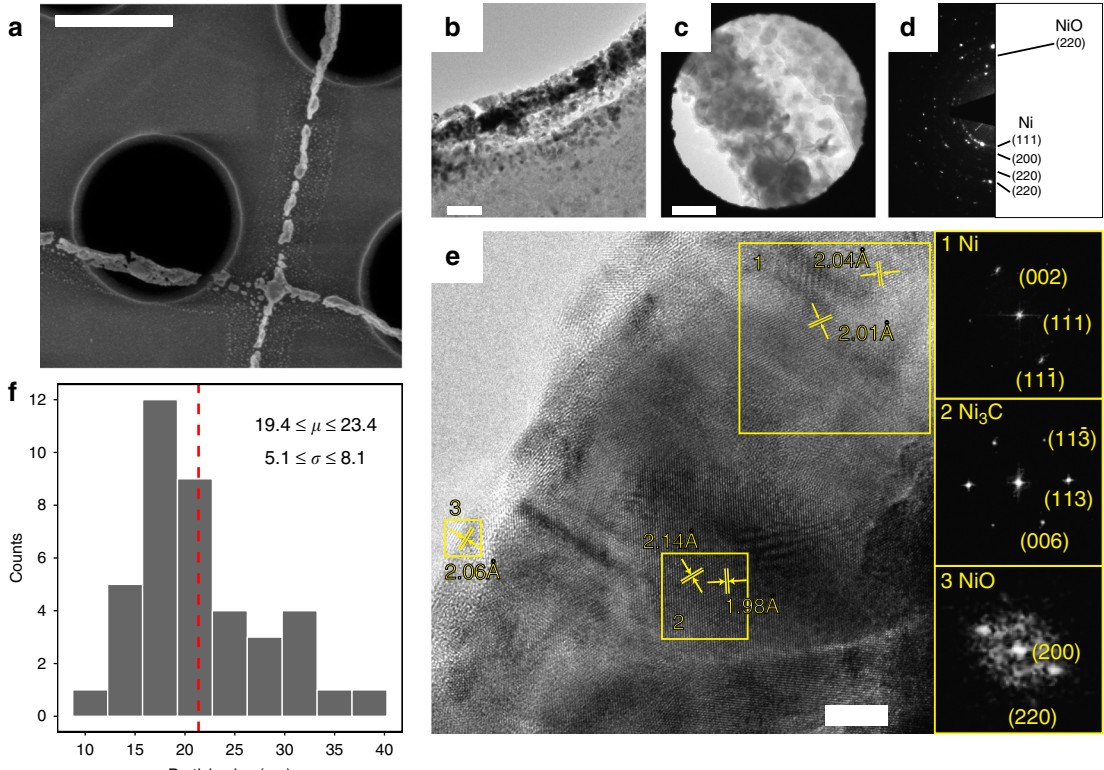

**Fig. 3** TEM characterization of the resulting metal structure. **a** SEM image of nickel beams fabricated directly on a 200-nm thick SiN membrane TEM grid **b** Low-magnification TEM of a 100 nm nickel beam overhanging the edge of 1.25-μm hole in a SiN membrane. **c** TEM image of the metal sample region where the diffraction pattern was taken. **d** Electron diffraction pattern shows that the printed beam consists mostly out of nanocrystalline nickel with a small amount of nickel oxide. **e** HRTEM image of a printed metal beam. Analysis of atomic plane distances using FFT shows predominantly nanocrystalline nickel (region 1) with some amount of nickel carbide in the interior (region 2) and nickel oxide at the surface (region 3). **f** Grain size histogram for $n = 40$ particles measured from a TEM image showing 95% confidence intervals for the mean grain size ($\mu$) and the standard deviation ($\sigma$) (see Supplementary Fig. 2 and Supplementary Table 1). Scale bars are 1 μm for **a**, 100 nm for **b**, 50 nm for **c**, and 5 nm for **e**

octahedral silver lattices with ~40-μm diameter beams[16] and ~2–7 times higher than the stainless steel lattices with ~200 μm diameter beams[37]. It appears to be on the same order as NiTi octahedral lattices with ~250-μm diameter beams[40] and AlSi10Mg diamond lattices with ~400 μm beams[41]. This suggests that the AM process developed in this work is capable of producing architectures with feature sizes that are an order of magnitude smaller than those fabricated using existing AM processes while retaining high strength. The specific strength calculations were performed with the assumption of monolithic beams, which leads to its underestimation because the nanocrystalline Ni within the beams has 10–30% residual porosity.

Some of the existing literature on the deformation of nanoporous metallic foams[42] and individual metallic nano-pillars[10,43] report higher strengths upon uniaxial compression than ones reported in this work. The key difference between the strength reported in this work and those in previous reports is that it is representative of the structural strength of the nanolattice, where each beam has heterogeneous porous microstructure, as well as each nodal junction, and both are subjected to a complex stress state upon global compression. The microstructure that comprises nanolattices in this work is nanocrystalline and nanoporous, and has different levels of hierarchy in the sense that each individual beam is nanocrystalline and nanoporous, as well as the entire structure. This microstructure within the individual beams stems from sintering of the Ni nanoparticles after the organic components volatilize; it's in distinct contrast to the monolithic metallic beams in all other literature on the deformation of nanoporous materials. This microstructure is detrimental to the overall

structural strength in two ways: (1) the additional porosity within each beam lowers the overall relative density of the architecture, and (2) upon mechanical deformation, each sintered junction experiences a local stress state, which creates an effective stress concentration in the material at an adjacent pore. The pores that border these regions of local stress concentrations can be viewed as notches or flaws that serve as locations of failure initiation upon mechanical loads. The distribution of nano-pores in each beam that comprises the nanolattices in this work leads to a distribution in the local failure strengths, which—in combination with the detrimental effects of lower relative density and the presence of junctions—serves to lower the overall structural strength.

The specific strength of the Ni nano-lattices in this work is 50–80% lower than that of Cu meso-lattices with a similar relative density reported in ref. [11], which likely stems from the lattice strength being governed by that of the monolithic, fully dense beams with grains spanning full beam width. The strength of nanoporous Au stochastic foams in ref. [42] was reported to be close to that of monolithic gold because each ligament is a virtually defect-free, single crystalline beam, whose strength approaches ideal strength of gold[42]. These foams have a fundamentally different microstructure compared to the nanolattices in this work in that they are stochastic foams with relatively slender, curved single-crystalline pristine beams. A direct comparison between the compressive strengths of nanocrystalline Ni nanolattices in this work and those of hollow lattices reported in refs [12,14,15,32,44] may be misleading, because this work is focused on solid-beam metallic nanolattices, which deform via compression and plastic flow upon uniaxial compression; the others

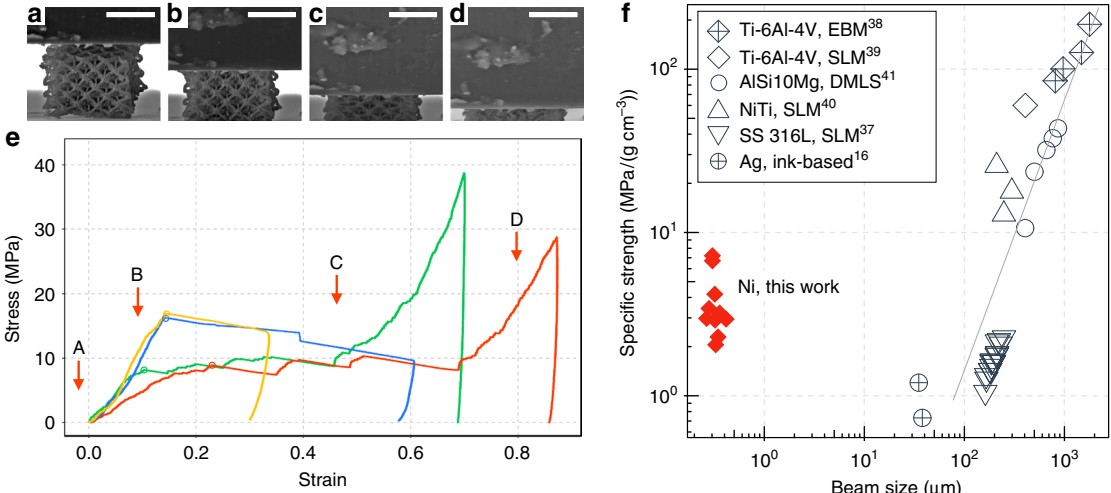

**Fig. 4** In situ uniaxial compression of 3D printed nickel octet nanolattices. **a–d** SEM images of the nickel structure during the compression test **a** before full contact, **b** in the elastic regime, **c** during layer-by-layer collapse, and **d** during densification. **e** Stress–strain data for compression of four nickel nanolattices. Letters on the graph correspond to the regions represented by **a–d**. **f** Specific strength-beam size plot showing properties of nickel nanolattices compared to other metal lattices fabricated using selective laser melting (SLM), direct metal laser sintering (DMLS), electron beam melting (EBM), and ink-based methods. See Supplementary Table 2 for data and references. Scale bars are 5 μm for **a–d**

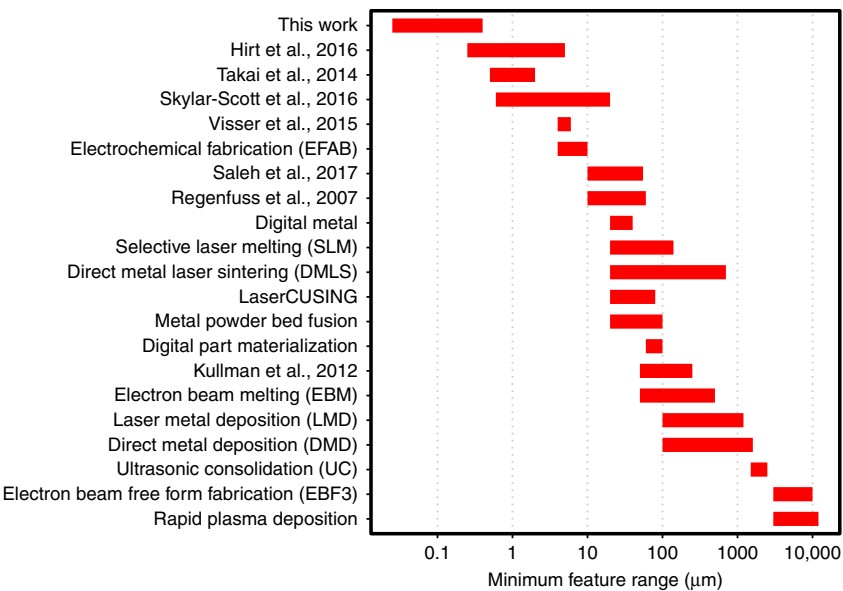

**Fig. 5** Comparison of minimum feature sizes for commercial and potentially scalable metal additive manufacturing technologies. Using metal-containing photoresist allows to fabricate complex 3D geometries with the resolution that is an order of magnitude finer than that of the state-of-the-art metal AM methods. See Supplementary Table 3 for data and references

contain hollow shell beams and undergo a different deformation mechanism upon compression that includes shell buckling and layer-by-layer collapse.

Figure 5 shows minimal reported printed feature sizes demonstrated in this work and many other metal AM processes available today (see Supplementary Table 3). The plotted ranges include both layer thickness and minimum lateral feature size. The minimum z-feature is determined by the thickness of a single layer of material. The minimum lateral feature is defined by multiple factors, which include the energy beam spot size and control over the melt pool. The data in Fig. 5 demonstrates that the AM process developed in this work is capable of producing features that are an order of magnitude smaller compared to those produced by other 3D-capable AM processes.

Another key aspect of any metal AM process is the throughput. Using hybrid organic–inorganic photoresist developed in this work allows for writing speeds of 4–6 mm s$^{-1}$, which is ~100 times faster than that for TPL of metal salts[24]. Comparing the speeds of metal AM processes with different resolutions can be accomplished by normalizing the write speed (μm s$^{-1}$) by the feature size (μm) or by normalizing the volumetric throughput (μm$^3$ s$^{-1}$) by the voxel volume (μm$^3$) (see ref. [7]). For a typical 300–600 nm feature size printed by TPL[45], writing speeds in this work correspond to defining 6700–20000 voxels s$^{-1}$, a printing speed that is out of reach for state-of-the-art micro-scale metal AM techniques, i.e., electrohydrodynamic printing (0.05–300 voxel s$^{-1}$), local electroplating (0.04–1.0 voxels s$^{-1}$), focused beam methods (0.01–0.8 voxels s$^{-1}$), and direct ink writing

(0.7–3000 voxels s$^{-1}$)[7] (see Supplementary Table 4 for linear writing speed and volumetric throughput data). High scanning speeds and intrinsic advantage of parallelizing light delivery using lithographic methods suggest that the presented AM process lends itself to streamlined and efficient manufacturing of metal nano-architectures.

We developed an AM process to create 3D nano-architected metals using an efficient lithography-based approach. Using this process, we fabricated Ni octet-lattices with 2-μm unit cells, 300–400-nm beams and 30-nm layers. The resolution of this method allows printing metal features with 25–100-nm dimensions, which is an order of magnitude smaller than feature sizes produced using all other 3D-capable metal AM methods without sacrifice in mechanical strength. This nanoscale metal AM method is not limited to nickel: other organometallics can be used to derive UV-curable metal-based photoresists using similar chemical synthesis. Nanoscale AM of metals has direct implications and opportunities for streamlined production of complex sub-millimeter devices, including 3D MEMS[6], 3D microbattery electrodes[46], and microrobots and tools for minimally invasive medical procedures[47].

## Methods

**UV-curable metal-based photoresist**. Acrylic acid (anhydrous, 99%), propylene glycol monomethyl ether acetate (PGMEA) (>99.5%), dichloromethane (anhydrous, ≥99.8%), 2-methoxyethanol (anhydrous, 99.8%), and isopropyl alcohol (IPA) (99.7%) were purchased from Sigma Aldrich. Nickel 2-methoxyethoxide, 5% w/v in 2-methoxyethanol was purchased from Alfa Aesar, and 7-diethylamino-3-thenoylcoumarin was purchased from Exciton. Acrylic acid (100 mg) was slowly added to nickel 2-methoxyethoxide solution (1290 mg) in a glove box and manually agitated. We observed the nearly immediate change of the solution color from brown to green, which is indicative of a ligand exchange reaction[48]. The mixture was held at low pressure in the antechamber of the glove box for 45 min to remove ~60% of 2-methoxyethanol. The resulting precursor was then taken out of the glove box, mixed with 300 mg of pentaerythritol triacrylate, and agitated using a vortex mixer for 1 min. 7-diethylamino-3-thenoylcoumarin (23 mg) was dissolved in 100 mg of dichloromethane, added to the mixture, which was then agitated using a vortex mixer for 1 min.

**Two-photon lithography**. Metal-containing polymer structures were fabricated on a silicon chip (1 × 1 cm) using a commercially available TPL system (Photonic Professional GT, Nanoscribe GmbH). Metal-containing photoresist was drop cast on a glass slide (0.17 mm thick, 30 mm in diameter) and confined between the glass slide and a silicon chip using 100-μm thick, 2 × 10-mm ribbons of Kapton tape as spacers. Laser power and scan speeds were set at at 17.5–22.5 mW and 4–6 mm s$^{-1}$, respectively. After the printing process, the samples were developed in 2-methoxyethanol for 1 h, followed by immersion in PGMEA for 10 min and filtered IPA for 5 min. The samples were then processed in a critical point dryer (Autosamdri-931).

**Pyrolysis**. Pyrolysis of the cross-linked metal-containing structures was conducted in two steps in a quartz tube furnace using 4″ quartz tube. As the first step, a heating profile of 2 °C min$^{-1}$ to 1000 °C, hold at 1000 °C for 1 h was applied under 1 L min$^{-1}$ argon flow, and the part was let to cool down in the furnace to room temperature. During the second step the part was heated at 2 °C min$^{-1}$ to 600 °C under 1 L min$^{-1}$ forming gas flow (5% H$_2$, 95% N$_2$), held at 600 °C for 1 h, and let to cool down to room temperature. No additional processing was performed after pyrolysis.

**Materials characterization**. SEM images were obtained using an FEI Versa 3D DualBeam. SEM EDS characterization was performed using a Zeiss 1550VP FESEM equipped with an Oxford X-Max SDD system using 10 kV electron beam.

TEM and TEM EDS were performed using FEI Tecnai F30ST (300 kV) transmission electron microscope equipped with Oxford ultra-thin window EDS detector. TEM sample was prepared by fabricating metal structures directly on PELCO Holey Silicon Nitride Support Film for TEM with 1250-nm holes (Ted Pella) following the process described above.

Phases and Miller indices for the phases in HRTEM image (Fig. 3e) were assigned based on the two lattice distances $d_{hkl}$ and the angle measured from FFT patterns within the outlined regions. Representative regions 1, 2, and 3 for the FFT analysis were chosen to contain a single particle or a region within a particle of interest. First, lattice distances $d_{hkl}$ for nickel, nickel (II) oxide, and nickel carbide were calculated based on the lattice parameters obtained from refs. [49–52]. The measured distances were then compared to the calculated values and matched

within 5% error. The phase assignment was verified by comparing the angle measured from the FFT pattern with the theoretical value for the obtained orientation, and further corroborated using the electron diffraction pattern in Fig. 3d.

**Particle size**. Particle sizes (see Supplementary Table 1) were measured from a bright-field TEM image using ImageJ (see Supplementary Fig. 2). Confidence interval on the mean particle size was calculated assuming normal distribution of the particle sizes and unknown variance using t-distribution ($n = 40$, $\alpha = 0.05$). Confidence interval on the variance of the particle size was calculated using $\chi^2$ distribution ($n = 40$, $\alpha = 0.05$).

**Mechanical characterization**. Uniaxial compression experiments were conducted using in situ nanomechanical instrument, SEMentor (InSEM; Nanomechanics and FEI Quanta 200). Samples were compressed using a diamond flat punch tip with a diameter of 170 μm at a constant strain rate of $10^{-3}$ s$^{-1}$. Relative density of each of the structures was calculated using a CAD model created in Abaqus with average unit cell sizes and beam diameters measured from the SEM images assuming fully-dense beams. Real-time deformation video and the mechanical data were simultaneously captured during the experiment (see Supplementary Movie 1).

Specific strength values shown in Supplementary Table 2 were calculated as the lattice strength divided by the lattice density. Lattice density values were taken from refs. [16,37,39] as reported. For structures in refs. [38,40,41], the lattice density was estimated as a product of the material density and the relative density of the structure. The material density of SLM NiTi was provided in ref. [40]. The material density of EBM Ti-6Al-4V in ref. [38] was assumed to be similar to Grade 5 Ti-6Al-4V, 4.43 g cm$^{-3}$, the closest material to Arcam Ti-6Al-4V ELI used in that work[53]. The material density of DMLS AlSi10Mg in ref. [41], 267 g cm$^{-3}$, was taken from the material datasheet[54]. Beam diameter values in refs. [16,37,38,40] were taken as reported. Beam diameters for AlSi10Mg lattices were estimated from Fig. 10 and 12 in ref. [41]. Beam diameters for SLM Ti-6Al-4V were measured from Fig. 2 in ref. [39]. For electroplated copper meso-lattices in ref. [11] specific strengths were calculated assuming bulk copper density of 8.96 g cm$^{-3}$.

**Data availability**. The data that support the findings of this study are available from the corresponding author upon reasonable request.

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

## Acknowledgements

The authors gratefully acknowledge the financial support of JRG's Vannevar-Bush Faculty Fellowship through the Department of Defense.

## Author contributions

A.V., S.D. and J.R.G. conceived the concept. A.V. performed synthesis, fabrication, SEM and EDS characterization. S.D. provided information on how to prepare the photoresist. A.K. performed TEM characterization and analysis. X.Z. and C.M.P. performed nano-compression experiments and analyzed the results. A.V. and J.R.G. wrote the manuscript. All authors commented on the manuscript. J.R.G. supervised the project.

## Additional information

**Competing interests:** The authors declare no competing financial interests.

