## [Peer Review File · Nature Communications]

Reviewers' Comments:

Reviewer #1:

Remarks to the Author:

The manuscript on "Additive Manufacturing of 3D Nano-Architected Metals" by Vyatskikh et al. describes the two-photon lithography and subsequent pyrolysis of a metal-rich photo-curable polymer resin in the shape of octet lattices. The structure and composition of the lattices are characterized using EDS and TEM, and the compressive properties of the lattices are evaluated inside of a SEM. Researchers should be able to reproduce the work given the level of detail provided in the manuscript. This manuscript presents an interesting fabrication process with novel elements (new photoresist chemistry and pyrolysis) that addresses an important challenge in the 3D printing community - the manufacture of 3D metallic structures with sub-micron features, but suffers from major flaws in analysis. I recommend publication in Nature Communications if the following major and minor issues can be addressed.

Major issues:

1. I find it very odd that previous reports from the Greer group and others on the fabrication and mechanical properties of nano and micro-lattices are not cited in this manuscript (Schaedler et al., Science; Jang et al., Nature Materials; Meza et al., Science; etc.), especially those that discuss the fabrication and mechanical properties of metallic lattices (Schaedler et al., Science; Gu and Greer, Extreme Mechanics Letters (2015); Lee et al., Nano Letters (2015)). The current work must be placed in the context of these previous studies with regard to mechanical properties (fig. 4f) and lattice beam size (fig. 5) to evaluate its novelty and achievements. I believe that the specific yield strength reported here is lower than that reported in previous studies on metallic nano-lattices, and that the statement that metallic lattices with slender beams have lower strength is untrue (line 152-153).
2. It does not make sense to extrapolate the specific strength of the cited macroscopic samples to smaller length scales to obtain the "expected" nano-lattice specific strength in figure 4f. Due to the differences in microstructure, structural geometry and composition between the large and small lattices, there is no reason to expect the mechanical properties of the small lattices to be similar to those of the large lattices. Again, a much better comparison would be to previous studies on nano and micro-lattices with the same octet geometry and similar composition.
3. The authors state that the porosity of their nanolattices is detrimental to mechanical strength (line 156-158). This is counter to the large body of literature on nanoporous metallic foams (Biener et al., Nano Letters (2006) among many others), and mechanical size effects at small scales (Greer and De Hosson, (2011)). The authors must discuss the microstructural differences between their work and previous studies that lead to the observed differences in mechanical properties to support their views.
4. Alternative, is the specific strength evaluated based on the density of the porous lattice structure, or an equivalent non-porous structure? If the density of the non-porous structure is used, this would obviously lead to the wrong conclusions about the effect of porosity.
5. Is the vertical spring support removed before mechanical testing? If not, how does it factor into the measured mechanical response of the lattice?
6. Mechanical properties are only measured on four samples. Given the large variation in mechanical response, and the alleged ease of fabricating these lattices, a larger number of tests in line with norms in the field of nano-mechanical testing should be performed to determine the source of this variation (differences in sample structure, or differences in testing conditions).

Minor issues:

7. Figure 3f. Explain the meaning of mu and sigma in the figure caption.

8. What is the total processing time (lithography and pyrolysis) to make these structures? How does this compare to other additive manufacturing techniques?

9. Was the TEM analysis of elemental composition in figure 3e performed on a single particle or multiple particles? This should be included in the text.

Writing and organization:

10. The quality of writing should be improved. For example, lines 39-43 is very confusing, written in a different verb tense as the rest of the paragraph, and possibly grammatically incorrect.

11. The conclusion section is labeled as the discussion section. The manuscript should be re-structured to have an actual discussion section.

12. Line 406: characterization is written twice

13. Citations should be included in the text as "ref. #" when referred to directly. Example: lines 258 to 266.

Reviewer #2:

Remarks to the Author:

General comments:

Overall this is a very good manuscript which is worthy of publication in Nature Communications. The authors report on a new way to achieve metallic nanoscale architectures using additive manufacturing and advanced resin chemistries. These chemistries allow for the production of nanolattices in a two-photon lithography system and represent the key contribution of the paper. This was achieved through incorporating a Ni-based MOF into a photo-resin, fabrication with a NanoScribe two-photon system, and subsequent thermal post-processing in the form of pyrolyzation of the polymer leaving only the Ni. The process was successful in yielding nanolattices primarily made of Ni. This is an interesting and useful result as it is a more direct way to generate metallic nanostructures which have been shown to be extremely interesting by the corresponding author's group in prior papers as well as by other reputable groups around the world. While the work is generally excellent, I have two overall requests/suggestions which the authors should address prior to publication, as well as some detailed comments:

- The claims regarding scalability which appear throughout the manuscript seem to be a bit of an over-reach. Two-photon lithography, in its current form, has not been scaled effectively. While the write speeds discussed in this paper are fast, it is still very difficult to build substantial structure with a NanoScribe and this needs to be acknowledged. Write speed is not the only requirement for scalability – overall build volume is still very small in this process and the smaller the feature size, the longer it takes to build substantial structure. The authors use "voxels per second" as a metric but this is deceptive as all processes have different voxel sizes and this would over-state the speed/scalability of this process. While there are some groups working on scale-up of the process, it has not yet been realized.
- The resultant mechanical properties are very good due to the very fine ~20nm nanocrystalline structure. However, the paper lacks some in-depth discussion of the deformation mechanism. Particularly, how does the very long plateau shown in the data come about? Is this plateau a result of the residual high porosity of 10-30%? Some discussion clarifying this would be useful.

Detailed comments:

1. Line 25-26: Authors claim, “scalable pathway” to nanolattices – this is a bit overstated. Two-photon has not been shown to be particularly scalable and this work does not address this.
2. Line 32-33: While I generally agree with the authors claim that “no established method is available for printing 3D features below these dimensions” it is notable that they do not compare this method to those where nanoscale coatings and/or nanoparticle suspensions have been used. This includes work by the corresponding author’s group. I believe that the work in this paper shows a superior method to achieve metallic lattices, so this comparison should be favorable.
3. Line 35-37: I agree with the statement “Accessing these phenomena requires developing a process to fabricate 3D metallic architectures with macroscopic overall dimensions and individual constituents in the sub-micron regime.” However, as previously stated regarding scalability, I’m not sure this paper addresses the “macroscopic” part of this.
4. Line 56: “We developed a scalable...” – same comment regarding scalability. Suggest softening these statements.
5. Line 73: “~80% shrinkage” – is this isotropic (assume it is but maybe state this) and is this volumetric?
6. Line 92: Composition is reported as 91.8% Ni, 5.0% O, 3.2% C – is this purity level good or bad? Can it be better? Does it impact properties?
7. Line 139: “None of the nanolattices recovered after deformation.” Would you expect them to recover if strut diameters begin to approach the grain size?
8. Line 156-158: Reports “10-30% residual porosity..” What is impact of this on mechanical properties? Have you considered something like a hot isostatic press to remove porosity?
9. Line 166-175: This section again discusses write speeds and scalability. The authors use “voxels/sec” as a metric of comparison to other printing methods. However, they do not define “voxel” very well. Is a voxel the same volume for all processes? I don’t think it is. Why not use actual volume of fabricated material per second? This would be more appropriate. The NanoScribe generally has a very small voxel element so using this as a metric will overstate its scalability. Again, I believe these claims are bit over-stated. A more transparent metric should be used.

Comment 1

Reviewer 1:

1. I find it very odd that previous reports from the Greer group and others on the fabrication and mechanical properties of nano and micro-lattices are not cited in this manuscript (Schaedler et al., Science; Jang et al., Nature Materials; Meza et al., Science; etc.), especially those that discuss the fabrication and mechanical properties of metallic lattices (Schaedler et al., Science; Gu and Greer, Extreme Mechanics Letters (2015); Lee et al., Nano Letters (2015)). The current work must be placed in the context of these previous studies with regard to mechanical properties (fig. 4f) and lattice beam size (fig. 5) to evaluate its novelty and achievements. I believe that the specific yield strength reported here is lower than that reported in previous studies on metallic nano-lattices, and that the statement that metallic lattices with slender beams have lower strength is untrue (line 152-153).

Response: We thank the reviewer for the suggestion to include the specified references into the manuscript. We have now included references to Schaedler et al., Science (2011); Jang et al., Nature Materials (2013); Meza et al., Science (2014); Gu and Greer, Extreme Mechanics Letters (2015); Lee et al., Nano Letters (2015); Montemayor and Greer, Journal of Applied Mechanics (2015); and Lontas and Greer, Acta Materialia (2017) in the revised manuscript. We chose to exclude references to the works previously published by Greer Group in Fig. 4f because its aim is to convey the specific strengths of metallic 3D architectures fabricated using only metal additive manufacturing processes. The references above describe either metal structures with monolithic beams fabricated by electroplating into a template (Gu and Greer, Extreme Mechanics Letters (2015)) or shell-beam architectures made by coating a template with a layer of metal (Schaedler et al., Science (2011); Lee et al., Nano Letters (2015); Montemayor and Greer, Journal of Applied Mechanics (2015); and Lontas and Greer, Acta Materialia (2017)), which are not metal AM processes. Similarly, Fig. 5 focuses on the resolution of the existing metal AM processes.

To address the reviewer's critique, we have included the specific strength of the solid-beam Cu lattices reported by Gu and Greer in the discussion section of the revised manuscript and in the revised Supplementary Table 2. A direct comparison between the compressive strengths of nanocrystalline Ni nanolattices in this work with those of the metallic lattices reported by Montemayor and Greer, Schaedler et al., Lee et al., and Lontas and Greer may be misleading because this work is focused on solid-beam metallic nanolattices, which deform via compression and plastic flow upon uniaxial compression; all others contain hollow shell beams and undergo a different deformation mechanism upon compression that includes shell buckling and layer-by-layer collapse.

Comment 2

Reviewer 1:

2. It does not make sense to extrapolate the specific strength of the cited macroscopic samples to smaller length scales to obtain the “expected” nano-lattice specific strength in figure 4f. Due to the differences in microstructure, structural geometry and composition between the large and small lattices, there is no reason to expect the mechanical properties of the small lattices to be similar to those of the large lattices. Again, a much better comparison would be to previous studies on nano and micro-lattices with the same octet geometry and similar composition.

Response: We thank the reviewer for this comment. It may indeed be misleading to extrapolate the specific strength plot to the micron-sized beam dimensions precisely for the reasons that the reviewer brings up: among others, the differences in microstructure, composition, and flaw distribution, may lead to drastically different mechanical performance at smaller sizes. We have now revised Figure 4 accordingly and included a comparison to the compressive strength of the structures with monolithic Cu beams fabricated by electroplating into a template (Gu and Greer, *Extreme Mechanics Letters* (2015)) in the discussion section of the manuscript.

Comment 3

Reviewer 1:

3. The authors state that the porosity of their nanolattices is detrimental to mechanical strength (line 156-158). This is counter to the large body of literature on nanoporous metallic foams (Biener et al., Nano Letters (2006) among many others), and mechanical size effects at small scales (Greer and De Hosson, (2011)). The authors must discuss the microstructural differences between their work and previous studies that lead to the observed differences in mechanical properties to support their views.

Response: It is true that some of the existing literature, including the publications on the deformation of nanoporous metallic foams and of the individual metallic nano-pillars brought up by the reviewer, report higher strengths upon uniaxial compression (Biener's and Volkert's groups). The key difference between the strength reported in this work and previous reports is that it is representative of the **structural** strength of the nanolattice, where each beam has heterogeneous porous microstructure, as well as each nodal junction, and both are subjected to a complex stress state upon global compression. The microstructure that comprises nanolattices in this work is nanocrystalline and nanoporous, and has different levels of hierarchy in the sense that each individual beam is nanocrystalline and nanoporous, as well as the entire structure. This microstructure within the individual beams stems from sintering of the Ni nanoparticles after the organic components volatilize; it is in distinct contrast to the monolithic metallic beams in all other literature on the deformation of nanoporous materials. This microstructure is detrimental to the overall structural strength in two ways: (1) the additional porosity within each beam lowers the overall relative density of the architecture and (2) upon mechanical deformation, each sintered junction experiences a local stress state, which creates an effective stress concentration in the material at an adjacent pore. The pores that border these regions of local stress concentrations can be viewed as "notches" or "flaws" that serve as locations of failure initiation upon mechanical loads. The distribution of nano-pores in each beam that comprises the nanolattices in this work leads to a distribution in the local failure strengths, which – in combination with the detrimental effects of lower relative density and the presence of junctions – serves to lower the overall structural strength.

In all the nano- and micro-pillar studies, the individual nano structure was subjected to the state of (nearly) uniaxial compression, with the competing effects of intrinsic and extrinsic length scales contributing to the overall deformation and strength (see Greer and De Hosson, (2011)). For example, the majority of single crystalline metallic nano- and micro-pillars exhibit the so-called "smaller is stronger" size effect in a power law fashion, which stems from the competing rates of dislocation annihilation and dislocation nucleation. Nanocrystalline metallic nano-pillars have been shown to become weaker with size reduction (Gu et al., Nano Letters (2012), Jang and Greer, Scripta Materialia (2011), Yang, B.; et al Philos. Mag. (2012)) because of the activation of grain boundary deformation and sliding, particularly in the grains that are adjacent to the free surfaces, in response to uniaxial deformation. To the best of the authors' knowledge, no in-depth literature exists on the deformation mechanisms in individual nanoporous nanocrystalline nano- or micro-pillars.

Literature on the deformation of nanoporous metals – for example, the work of Biener, Hodge, Volkert, et al Nano Letters 2006 - reveals a close-to- $\mu/10$ compressive strength of nanoporous gold cylinders with micron-sized diameters and 10 nm-sized individual ligands. Their overall strength is reported to be close to that of monolithic gold because each ligament is a virtually defect-free, single crystalline beam, whose strength approaches ideal strength of gold. These foams have a fundamentally different microstructure compared to the nanolattices in this work in that they are stochastic foams with relatively slender, curved single-crystalline pristine beams. The overall structural strength is governed by the strength of each ligand (reported to be 4.6 GPa), which is close to the theoretical shear stress for gold (4.3 GPa).

These foams have a single level of hierarchy in that the beams that comprise the porous architecture are homogeneous. The nanolattices in this work have multiple levels of hierarchy: the beams that

comprise the micron-level architecture have complex microstructure that is nanocrystalline and nanoporous; with short sintered necks between nanoparticles and no long or slender sub-beams within it. Upon mechanical deformation, each individual beam and each junction experiences a distribution of local stresses within it, which create a landscape of stress concentrations around “flaws” and lead to failure at lower applied stresses. The importance and the significant contribution of the nodes to the deformation and the lower-than-expected scaling of stiffness (and strength) with relative density was recently published in Meza et al., *Acta Materialia* (2017). The above discussion has now been included in the revised manuscript.

Comment 4

Reviewer 1:

4. Alternative, is the specific strength evaluated based on the density of the porous lattice structure, or an equivalent non-porous structure? If the density of the non-porous structure is used, this would obviously lead to the wrong conclusions about the effect of porosity.

Response: This is a very important distinction. It would be incorrect to only account for the density of the prescribed architecture because the density of the “parent solid” is not that of the monolithic, non-porous nickel. We evaluated the specific strength as the measured strength of the lattice divided by the lattice density, which is a product of the relative density of the structure and the material density. For nickel nano-lattices fabricated in this work the relative density was estimated using a CAD model with average unit cell sizes and beam diameters measured from the SEM images assuming fully-dense beams. This assumption means that the calculated values are lower-bound estimates of the specific strength. We estimated the porosity of the nanoporous nickel that comprises each beam to be ~ 10-30% using SEM and TEM images. In our ongoing current work on this project, we are exploring ways to accurately quantify the beam porosity.

Comment 5

Reviewer 1:

5. Is the vertical spring support removed before mechanical testing? If not, how does it factor into the measured mechanical response of the lattice?

Response: The four corner springs in the support structure were no longer connecting the sample to the substrate after pyrolysis, as can be seen in Fig. 4a, and did not contribute to the mechanical response of the lattice. The central pillar in the support structure failed after the initial contact, which allowed for establishing full contact between the sample and the substrate and for sample alignment. This contributed to the mechanical response of the sample in the toe region, which was not included in the compression data for the compression strength calculation.

Comment 6

Reviewer 1:

6. Mechanical properties are only measured on four samples. Given the large variation in mechanical response, and the alleged ease of fabricating these lattices, a larger number of tests in line with norms in the field of nano-mechanical testing should be performed to determine the source of this variation (differences in sample structure, or differences in testing conditions).

Response: To comply with the reviewer's request, we have now fabricated six additional samples following the AM process described in the manuscript and compressed them using the same methodology. The additional stress-strain data is provided in Supplementary Fig. S4 and conveys its close resemblance to the original data shown in Figure 4e. Data for additional samples was added to Supplementary Table S2 and to Fig. 4f. We chose not to modify Figure 4e in the main manuscript because it shows representative stress-strain data for four samples out of ten, and it would crowd the plot. We revised the language in the manuscript to reflect this change.

Comment 7

Reviewer 1:

7. Figure 3f. Explain the meaning of mu and sigma in the figure caption.

Response: We have modified the figure caption to read as follows:

Fig. 3 TEM characterization of the nano-architected Ni. **a** SEM image of the nickel beams fabricated directly on a 200 nm-thick SiN membrane within the TEM grid **b**. Low-magnification TEM image of a 100 nm-diameter nickel beam overhanging the edge of a 1.25 μm -diameter hole in the SiN membrane. **c** TEM image of the metal sample in the region where the diffraction pattern was obtained. **d** Electron diffraction pattern shows that the printed beam consists mostly out of polycrystalline nickel with a small amount of nickel oxide. **e** HRTEM image of a printed metal beam. Analysis of atomic plane distances using FFT shows predominantly polycrystalline nickel (region 1) with some amount of nickel carbide within the beams (region 2) and nickel oxide at the surface (region 3). **f** Grain size histogram for $n=40$ particles measured from a TEM image showing 95% confidence intervals for the mean grain size (μ) and the standard deviation (σ) (see Figure S2 and Supplementary Table S1)

Comment 8

Reviewer 1:

8. What is the total processing time (lithography and pyrolysis) to make these structures? How does this compare to other additive manufacturing techniques?

Response: The lithography time to fabricate a single structure shown in Fig. 1f is ~1 hr, and the total pyrolysis time is ~20 hr. We compare the throughput of the developed process to other state-of-the-art micro-scale AM processes in the Discussion section of the manuscript:

For a typical 300-600 nm feature size printed by TPL³⁵, writing speeds in this work correspond to defining 6700 – 20000 voxels s⁻¹, a printing speed that is out of reach for state-of-the-art micro-scale metal AM techniques, i.e. electrohydrodynamic printing (0.05-300 voxels s⁻¹), local electroplating (0.04-1.0 voxels s⁻¹), focused beam methods (0.01-0.8 voxels s⁻¹), and direct ink writing (0.7-3000 voxels s⁻¹)⁷.

We have also included a comparison of linear writing speeds and volumetric throughputs of these processes in the manuscript (see Supplementary Table S4).

Comment 9

Reviewer 1:

9. Was the TEM analysis of elemental composition in figure 3e performed on a single particle or multiple particles? This should be included in the text.

Response: FFT patterns shown in Fig. 3e were taken from regions that include a single particle or a region within a particle. The diffraction pattern in Fig. 3d was collected from a conglomerate of particles shown in Fig. 3c. We have included a clarification in the Methods section regarding representative regions 1, 2, and 3 in Fig. 3e that were chosen to show the phases of the materials present:

Phases and Miller indices for the phases in HRTEM image (Fig. 3e) were assigned based on the two lattice distances d_{hkl} and the angle measured from FFT patterns within the outlined regions. Representative regions 1, 2, and 3 for the FFT analysis were chosen to include a single particle or a region within a particle of interest.

Comment 10

Reviewer 1:

10. The quality of writing should be improved. For example, lines 39-43 is very confusing, written in a different verb tense as the rest of the paragraph, and possibly grammatically incorrect.

Response: We have revised this paragraph accordingly:

Minimum feature size in metal AM is generally limited by the material feedstock, i.e. the method of supplying metal in powder, wire, sheet or ink form during fabrication. Inkjet-based methods^{12,13} manipulate 40-60 μm droplets of metal inks, limiting the smallest features to at least the size of a solidified droplet. Wire- and filament-based processes, such as Plasma Deposition⁴ and Electron Beam Freeform Fabrication (EBF3)¹⁴, rely on locally melting a >100 μm -diameter metal wire, which produces millimeter-sized features. Powder-based processes, such as Selective Laser Melting (SLM) and Laser Engineered Net Shaping (LENS)¹⁵, consolidate ~ 0.3 - 10 μm metal powder particles, which limits the smallest feature size to about 20 μm ^{6,16}. Overcoming these resolution limitations requires a capability to manipulate nanoscale quantities of metals in a stable and scalable 3D printing process.

Comment 11

Reviewer 1:

11. The conclusion section is labeled as the discussion section. The manuscript should be re-structured to have an actual discussion section.

Response: We thank the reviewer for the evaluation and feedback. We have re-structured the manuscript to include separate discussion and conclusion sections.

Comment 12

Reviewer 1:

12. Line 406: characterization is written twice

Response: We have corrected the figure caption accordingly.

Comment 13

Reviewer 1:

13. Citations should be included in the text as “ref. #” when referred to directly. Example: lines 258 to 266.

Response: We thank the reviewer for pointing this out. We have modified direct references accordingly.

Comment 14

Reviewer 2:

- The claims regarding scalability which appear throughout the manuscript seem to be a bit of an over-reach. Two-photon lithography, in its current form, has not been scaled effectively. While the write speeds discussed in this paper are fast, it is still very difficult to build substantial structure with a NanoScribe and this needs to be acknowledged. Write speed is not the only requirement for scalability – overall build volume is still very small in this process and the smaller the feature size, the longer it takes to build substantial structure. The authors use “voxels per second” as a metric but this is deceptive as all processes have different voxel sizes and this would over-state the speed/scalability of this process. While there are some groups working on scale-up of the process, it has not yet been realized.

Response: We thank the reviewer for this comment. We have adjusted statements regarding scalability throughout the manuscript. We agree with the reviewer that “the smaller the feature size, the longer it takes to build substantial structure”, which is the reason why the “voxel s⁻¹” metric was suggested previously to compare writing speeds between AM processes with different resolutions (Hirth et al., Adv. Mater., 2017). Please see a more detailed response about this in Comment 24.

Comment 15

Reviewer 2:

- The resultant mechanical properties are very good due to the very fine ~20nm nanocrystalline structure. However, the paper lacks some in-depth discussion of the deformation mechanism. Particularly, how does the very long plateau shown in the data come about? Is this plateau a result of the residual high porosity of 10-30%? Some discussion clarifying this would be useful.

Response: We thank the reviewer for the question. A detailed discussion on the deformation mechanisms as a function of microstructure in small-scale features and architectures has now been included in the revised manuscript. For a more detailed discussion, we refer the reviewer to the response to Comment 3.

Comment 16

Reviewer 2:

1. Line 25-26: Authors claim, “scalable pathway” to nanolattices – this is a bit overstated. Two-photon has not been shown to be particularly scalable and this work does not address this.

Response: We thank the reviewer for this comment. We have adjusted this statement accordingly.

Comment 17

Reviewer 2:

2. Line 32-33: While I generally agree with the authors claim that “no established method is available for printing 3D features below these dimensions” it is notable that they do not compare this method to those where nanoscale coatings and/or nanoparticle suspensions have been used. This includes work by the corresponding author's group. I believe that the work in this paper shows a superior method to achieve metallic lattices, so this comparison should be favorable.

Response: We thank the reviewer for this suggestion. In this manuscript, we focused on comparing the Additive Manufacturing method we developed to the previously reported metal additive manufacturing processes and metal AM-fabricated structures. Based on the reviewer's suggestion and scope of the manuscript, we have included a comparison of specific yield strength of the structures reported in this work to some other relevant literature, which includes solid beam metal mesolattices fabricated by electroplating process (Gu and Greer, *Extreme Mechanics Letters* (2015)). A direct comparison between the compressive strengths of nanocrystalline Ni nanolattices in this work with those of the shell-beam architectures made by coating a template with a layer of metal (Schaedler et al., *Science* (2011); Lee et al., *Nano Letters* (2015); Montemayor and Greer, *Journal of Applied Mechanics* (2015); Montemayor et al., *Advanced Engineering Materials* (2013); and Liontas and Greer, *Acta Materialia* (2017)) may be misleading because this work is focused on solid-beam metallic nanolattices, which deform via compression and plastic flow; shell-beam structures undergo a different deformation mechanism upon compression that includes shell buckling and layer-by-layer collapse.

Comment 18

Reviewer 2:

3. Line 35-37: I agree with the statement “Accessing these phenomena requires developing a process to fabricate 3D metallic architectures with macroscopic overall dimensions and individual constituents in the sub-micron regime.” However, as previously stated regarding scalability, I’m not sure this paper addresses the “macroscopic” part of this.

Response: We thank the reviewer for this comment. We have adjusted the statements regarding scalability throughout the manuscript.

Comment 19

Reviewer 2:

4. Line 56: “We developed a scalable...” – same comment regarding scalability. Suggest softening these statements.

Response: We thank the reviewer for this comment. We have adjusted this statement accordingly in the revised manuscript.

Comment 20

Reviewer 2:

5. Line 73: “~80% shrinkage” – is this isotropic (assume it is but maybe state this) and is this volumetric?

Response: The shrinkage is isotropic and results in ~80% smaller linear dimensions. We have added a clarifying sentence to the Results section.

Comment 21

Reviewer 2:

6. Line 92: Composition is reported as 91.8% Ni, 5.0% O, 3.2% C – is this purity level good or bad? Can it be better? Does it impact properties?

Response: This is a very important point. The SEM EDS analysis is generally not the most accurate method to evaluate carbon content and is highly dependent on carbon deposition in the SEM chamber (Donovan et al., Scanning Electron Microscopy, Chapter in *Characterization of Materials* (John Wiley and Sons, 2002)). We observe <1% by volume of Ni₃C in the TEM, which confirms the presence of carbon in the as-fabricated structure. It is reasonable to expect some traces of carbon in the final pyrolyzed structure due to high solubility of carbon in nickel at 1000°C (~0.1 wt%, Lander et al., *Journal of Applied Physics* 23, 1305 (1952)), which leads to carbon precipitation at nickel surface upon cooling down to room temperature.

The presence of the oxygen peak in the EDS signal is consistent with nickel oxidation in air, which leads to the formation of native oxide and full oxidation of surface particles smaller than 6 nm (Wang et al., *J Nanosci Nanotechnol.*, 2011). We expect this effect to be especially pronounced in nanoporous nickel beams fabricated in this work since these were exposed to air before SEM and TEM analysis. TEM analysis reveals that these elements – O and C – are bound to Ni and form small amounts of NiO and Ni₃C. Lower oxygen content can be achieved by decreasing the porosity of the structural elements. Carbon content can be reduced by using oxygen plasma after the pyrolysis is complete, but another reduction step would then be needed to decrease the amount of oxygen in the sample. It is likely that the ~5wt% of O and ~3wt% of C contribute negligibly to the mechanical properties of the overall structure.

Comment 22

Reviewer 2:

7. Line 139: “None of the nanolattices recovered after deformation.” Would you expect them to recover if strut diameters begin to approach the grain size?

Response: The question of whether the nanolattices would recover if the struts had a bamboo-like structure (with or without the pores), i.e. each grain spanned the entire diameter of the strut, is an interesting one to explore. The recovery in the elastic, pre-yield regime is governed by the slenderness ratio of the beams and the boundary conditions at the nodes. This type of elastic recovery would be relatively insensitive to the microstructure, depending on it only in terms of the modulus. Intuitively, we do not expect metallic nanolattices with bamboo microstructure within the beams to recover in the global post-elastic regime because the local yielding events will occur at various points within the structure, with dislocation- and grain boundary-driven deformation processes governing local plasticity in the grains within the struts. The deformation of mechanical behavior of copper mesolattices with strut diameters on the order of the grain size (albeit in the micron range) were previously explored in Gu and Greer, *Extreme Mechanics Letters* (2015) and showed no recovery after yielding. Our current efforts in this project are dedicated to a more in-depth analysis of the material microstructure and its effect on the mechanical deformation of the overall architecture and will be prepared in a separate manuscript.

Comment 23

Reviewer 2:

8. Line 156-158: Reports “10-30% residual porosity..” What is impact of this on mechanical properties? Have you considered something like a hot isostatic press to remove porosity?

Authors: This is a very good question. Please see our response to Comment 15 for a detailed discussion of the effect of porosity on the mechanical properties.

We thank the reviewer for suggesting using a hot isostatic press. Our current efforts are dedicated in exploring pathways to accurately quantify the porosity within individual beams and to assess its effect on the overall deformation and strength. Methods to reduce porosity are also being explored.

Comment 24

Reviewer 2:

9. Line 166-175: This section again discusses write speeds and scalability. The authors use “voxels/sec” as a metric of comparison to other printing methods. However, they do not define “voxel” very well. Is a voxel the same volume for all processes? I don’t think it is. Why not use actual volume of fabricated material per second? This would be more appropriate. The Nanoscribe generally has a very small voxel element so using this as a metric will overstate its scalability. Again, I believe these claims are bit over-stated. A more transparent metric should be used.

Response:

We thank the reviewer for this comment. The metric “voxels s^{-1} ” to compare the write speeds of metal AM processes with different resolutions has been proposed in a recent review of micro-scale metal AM processes (Hirt et al., *Adv. Mater.*, 2017). The reasoning behind using this metric is the correlation between the process resolution and the volumetric throughput. Writing with higher resolution requires larger number of voxels to define the same geometry, which translates into longer processing times and lower volumetric throughputs. Comparing the speed between processes with different resolutions can be accomplished by normalizing the write speed ($\mu\text{m } s^{-1}$) by the feature size (μm) or by normalizing the volumetric throughput ($\mu\text{m}^3 s^{-1}$) by the voxel volume (μm^3) (Hirt et al., *Adv. Mater.*, 2017). We have added a clarification regarding the metric definition into the Discussion part of the manuscript and adjusted the statements regarding scalability.

Another key aspect of any metal AM process is the throughput. Using hybrid organic-inorganic photoresist developed in this work allows for writing speeds of $4\text{-}6 \text{ mm } s^{-1}$, which is ~ 100 times faster than that for TPL of metal salts²⁰. Comparing the speed between processes with different resolutions can be accomplished by normalizing the write speed ($\mu\text{m } s^{-1}$) by the feature size (μm) or by normalizing the volumetric throughput ($\mu\text{m}^3 s^{-1}$) by the voxel volume (μm^3)⁷. For a typical $300\text{-}600 \text{ nm}$ feature size printed by TPL³⁵, writing speeds in this work correspond to defining $6700 - 20000 \text{ voxels } s^{-1}$, a printing speed that is out of reach for state-of-the-art micro-scale metal AM techniques, i.e. electrohydrodynamic printing ($0.05\text{-}300 \text{ voxels } s^{-1}$), local electroplating ($0.04\text{-}1.0 \text{ voxels } s^{-1}$), focused beam methods ($0.01\text{-}0.8 \text{ voxels } s^{-1}$), and direct ink writing ($0.7\text{-}3000 \text{ voxels } s^{-1}$)⁷. High scanning speeds and intrinsic advantage of parallelizing light delivery using lithographic methods suggest that the presented AM process lends itself to streamlined and efficient manufacturing of metal nano-architectures.

We have also added Supplementary Table 4, which shows a comparison between writing speeds and volumetric throughputs of state-of-the-art micro-scale metal AM methods:

Supplementary Table 4

Comparison of linear and volumetric throughputs of representative micro-scale metal additive manufacturing technologies (data adopted from ref.²⁶)

#	Technology	Material	Feature size, μm	Writing speed*	Ref.
1	Direct Ink Writing (DIW)	Ag	0.6-20	500-2000 $\mu\text{m s}^{-1}$	25
2	Electrohydrodynamic (EHD) Printing	Ag, Co, Cu	0.7-3.0	0.16-3.3 $\mu\text{m s}^{-1}$	27
3	Laser-Induced Forward Transfer (LIFT)	Au, Cu	4.0-6.0	3000 $\mu\text{m}^3 \text{s}^{-1}$	24
4	Focused Electron Beam Induced Deposition (FEBID)	Pt	0.15-0.23	0.0002-0.0009 $\mu\text{m}^3 \text{s}^{-1}$	28
5	Cryo-FEBID	Pt	0.022-0.31	10 $\mu\text{m}^3 \text{s}^{-1}$	29
6	Meniscus-confined electroplating	Cu	12.0-15.0	0.18-0.4 $\mu\text{m s}^{-1}$	30
7	Local electrophoretic deposition	Au	0.5-2.0	0.30-0.67 $\mu\text{m s}^{-1}$	22
8	This work	Ni	0.025-0.4	4000-6000 $\mu\text{m s}^{-1}$	

*Volumetric ($\mu\text{m}^3 \text{s}^{-1}$) or linear ($\mu\text{m s}^{-1}$) writing speed is given when available

Reviewers' Comments:

Reviewer #1:

Remarks to the Author:

This version of the manuscript is much improved, especially the discussion section. The additional mechanical tests also add value to the manuscript. I recommend publication after the following (new) issues are addressed:

Major issue:

Lines 24-26 in the abstract state that the specific strength of the structures is higher than other small truss structures making using metal AM. This directly contradicts Figure 4F, which shows that many AM fabricated trusses have higher specific strength. It also directly contradicts lines 68-70, which state that the Ni nanolattices have comparable strength to other metal lattices. I am ok with publication if the phrase on line 25-26 ("which is up to an order of magnitude higher than that of the smallest truss architectures...") is removed.

Minor issue:

Line 194: violalize – misspelling

Line 209 – missing an "of"

Reviewer #2:

Remarks to the Author:

The authors have done an excellent job of addressing my few concerns and this paper is surely ready for publication. I thank the authors for clarifying the comparison of write speeds among the various technologies. It is now a lot more clear and the normalization by feature size makes sense. Additionally, the new text expanding on the mechanical performance of the structures adds detail and value to the manuscript. Finally, the inclusion of additional references, primarily from the authors own work, is appropriate. I congratulate the authors on excellent work. Thank you.

Comment 1

Reviewer 1: Lines 24-26 in the abstract state that the specific strength of the structures is higher than other small truss structures making using metal AM. This directly contradicts Figure 4F, which shows that many AM fabricated trusses have higher specific strength. It also directly contradicts lines 68-70, which state that the Ni nanolattices have comparable strength to other metal lattices. I am ok with publication if the phrase on line 25-26 (“which is up to an order of magnitude higher than that of the smallest truss architectures...”) is removed.

Response: We have corrected the statement in the abstract accordingly.

Comment 2

Reviewer 1:

Line 194: violalize – misspelling

Response: We thank the reviewer for catching this. We have corrected the word accordingly.

Comment 3

Reviewer 1:

Line 209 – missing an “of”

Response: We thank the reviewer for noticing this. We have corrected the statement accordingly.